

# Cavitons and spontaneous hot flow anomalies in a hybrid-Vlasov global magnetospheric simulation

Xochitl Blanco-Cano[1], Markus Battarbee[2], Lucile Turc[2], Andrew P. Dimmock[3,8], Emilia K. J. Kilpua[2], Sanni Hoilijoki[4], Urs Ganse[2], David G. Sibeck[5], Paul A. Cassak[6], Robert C. Fear[7], Riku Jarvinen[8,9], Liisa Juusola[9,2], Yann Pfau-Kempf[2], Rami Vainio[10], and Minna Palmroth[2,9]

[1]Instituto de Geofisica, Universidad Nacional Autonoma de Mexico, Mexico
[2]Department of Physics, University of Helsinki, Helsinki, Finland
[3]Swedish Institute of Space Physics, Uppsala, Sweden
[4]Laboratory for Atmospheric and Space Physics, University of Colorado, Boulder, USA
[5]NASA Goddard Space Flight Center, Greenbelt, Maryland, USA
[6]Department of Physics and Astronomy, West Virginia University, Morgantown, West Virginia, USA
[7]Department of Physics & Astronomy, University of Southampton, Southampton, UK
[8]Department of Electronics and Nanoengineering, School of Electrical Engineering, Aalto University, Espoo, Finland
[9]Finnish Meteorological Institute, Helsinki, Finland
[10]Department of Physics and Astronomy, University of Turku, Turku, Finland

*Correspondence to:* Xochitl Blanco-Cano (xbc@geofisica.unam.mx)

**Abstract.** In this paper we present the first identification of foreshock cavitons and the formation of spontaneous hot flow anomalies (SHFAs) with the Vlasiator global magnetospheric hybrid-Vlasov simulation code. In agreement with previous studies we show that cavitons evolve into SHFAs. In the presented run, this occurs very near the bow shock. We report on SHFAs surviving the shock crossing into the downstream region, and show that the interaction of SHFAs with the bow shock
can lead to the formation of a magnetosheath cavity, previously identified in observations and simulations. We report on the first identification of long-term local weakening and erosion of the bow shock, associated with a region of increased foreshock SHFA and caviton formation, and repeated shock crossings by them. We show that SHFAs are linked to an increase in suprathermal particle pitch-angles spreads. The realistic length scales in our simulation allow us to present a statistical study of global caviton and SHFA size distributions, and their comparable size distributions support the theory that SHFAs are formed
from cavitons. Virtual spacecraft observations are shown to be in good agreement with observational studies.

**Keywords.** Interplanetary physics (Planetary bow shocks), Magnetospheric physics (Solar wind–magnetosphere interactions), Space plasma physics (Numerical simulation studies)

## 1 Introduction

The interaction of the supermagnetosonic solar wind flow with the Earth's magnetosphere leads to the formation of a bow
shock, where the solar wind is decelerated, deviated, compressed and heated. Many works in the past have focused on studying the bow shock and the kinetic processes taking place in it, such as Cluster mission results in Space Sci. Rev., 118 (2005); Omidi et al. (2005); Blanco-Cano et al. (2006); Burgess and Scholer (2013), and Palmroth et al. (2015). However, there are





still many open questions about how the solar wind is processed at the bow shock and in the regions upstream and downstream of it, known as the foreshock and magnetosheath, respectively. Understanding how the plasma is modified at the foreshock, bow shock and in the magnetosheath is very important in terms of fundamental collisionless plasma physics, and also because it is this shocked and processed plasma, not the pristine solar wind, which ultimately interacts with the Earth's magnetic field and sometimes leads to geomagnetic disturbances.

The structure of a collisionless shock depends on its strength, given by the upstream magnetosonic Mach number $M_{ms}$ and the plasma density compression ratio; on the geometry, given by $\theta_{Bn}$ (the angle between the shock normal and the upstream magnetic field); and on the plasma beta ($\beta$). Shocks are classified as quasi-parallel (quasi-perpendicular) when $\theta_{Bn} < 45° (\theta_{Bn} > 45°)$.

Earth's bow shock usually has magnetosonic and Alfvénic Mach numbers in the range $2 \leq M_{ms} \leq 7$ and $2.5 \leq M_A \leq 12$ (Winterhalter and Kivelson, 1988). Due to its curvature and the Parker spiral configuration of the interplanetary magnetic field (IMF), part of the Earth's bow shock typically has a quasi-parallel geometry while in other regions there exists a quasi-perpendicular shock. The foreshock is the region connected magnetically to the shock, i.e., upstream from the quasi-parallel region. The bow shock is supercritical, which means that part of the solar wind kinetic energy is dissipated by the reflection of incident solar wind ions. Thus, the foreshock is permeated by a variety of suprathermal ion distributions, and a variety of ultra low frequency (ULF) waves (Hoppe et al., 1981) driven by ion instabilities which grow by the interaction of the reflected ions with the solar wind core. A detailed description of wave modes in the foreshock can be found in the review papers of Eastwood et al. (2005) and Wilson (2016). In addition to the ULF waves, large-scale (sizes of the order of Earth radii) transient structures such as cavitons (Omidi, 2007; Blanco-Cano et al., 2009; Kajdič et al., 2010), hot flow anomalies (HFAs) (Schwartz et al., 1985; Schwartz, 1995), and spontaneous hot flow anomalies (SHFA) (Zhang et al., 2013; Omidi et al., 2013; Omidi et al., 2014b) are also found in the foreshock. While the origin and evolution of foreshock ion distributions and waves have been studied for several decades, the study of transient structures is more recent.

Cavitons are structures observed in the foreshock that are characterized by dips in the magnetic field magnitude $B$ and plasma density $n$, bounded by overshoots in these parameters (Blanco-Cano et al., 2009). The temperature inside the cavitons is similar to the value in the surrounding plasma. They are proposed to form when transverse and compressive ULF waves interact non-linearly (Omidi and Sibeck, 2007) and are hence surrounded by intense compressive ULF waves. Foreshock suprathermal ions can be accumulated inside their cores. When formed, cavitons are carried with the solar wind flow towards the bow shock. An extensive statistical analysis by Kajdič et al. (2013) using Cluster data shows that cavitons occur for a wide range of solar wind conditions upstream of the quasi-parallel bow shock regime, and on average they show decrements of between 0.2 and 0.9 for both $\delta B/B_{sw}$ and $\delta n/n_{sw}$.

HFAs are observed upstream of the bow shock and similar to cavitons are characterized by decreases in the magnetic field magnitude and plasma density, but they also show a notable increase in temperature (Schwartz et al., 1985; Schwartz, 1995). As a consequence, they have enhanced plasma $\beta$. Flow inside HFAs is strongly decelerated and deflected. Typical sizes of HFAs deduced from observations are approximately 2-3 Earth radii ($R_E$) (e.g., Facskó et al., 2009). The formation of an HFA needs an external perturbation in the solar wind, e.g., a current sheet interacting with a bow shock. In this paper we focus on





spontaneous HFAs (SHFAs) that are similar in their characteristics to HFAs, except typically smaller in size (e.g., Kajdič et al., 2017), but have a different formation mechanism. SFHAs form primarily due to inherent foreshock structures, namely cavitons discussed above. Numerical hybrid simulations have shown that cavitons evolve into SHFAs as they move closer to the bow shock (Omidi et al., 2013). The proposed formation mechanism for SHFAs includes multiple ion reflections between foreshock

cavitons and the bow shock (Omidi et al., 2013), as cavitons approach the shock, and ion trapping occurs in the cavitons.

Foreshock ULF waves propagate towards the Sun in the solar wind frame with phase speeds of the order of the Alfvén speed, i.e., much smaller than the solar wind speed. As a consequence, they are convected towards the shock by the supermagnetosonic solar wind flow. Several observational and simulation studies have shown that ULF waves evolve into non-linear structures as they approach the shock, becoming compressive shocklets and SLAMS (short large amplitude magnetic structures) (Schwartz

and Burgess, 1991), which in turn play an active role in the reformation of the quasi-parallel bow shock, in the variability of the density of reflected ions, and in the variability in shock heating and rippling (see for example Burgess, 1989; Meziane et al., 2001; Mazelle et al., 2003). As a consequence of ULF waves, shocklets and SLAMS merging into the shock, the quasi-parallel portion of the bow shock is far from being a single well defined surface, but instead forms a highly corrugated/rippled extended structure, where inhomogeneous heating and solar wind processing can take place (see, for example, Schwartz and Burgess,

1991; Omidi et al., 2005; Blanco-Cano et al., 2009).

It has been shown that cavitons and SHFAs are also convected by the solar wind towards the shock (Kajdič et al., 2013). In a recent study, Omidi et al. (2016) demonstrated that SHFAs can result in the formation of magnetosheath cavities (Katırcıoğlu et al., 2009) which are associated with decreases in plasma density, bulk velocity and magnetic field magnitude, and enhancements in temperature. However, we still know very little about how the arrival of cavitons and SHFAs can modify the bow

shock structure and the magnetosheath. It is expected that their arrival at the bow shock will impact its structure, contributing to the formation of shock irregularities. More specifically, it is expected that they may lead to decrements in the shock magnetic field magnitude due to the decreased field inside these structures. If the interplay of shock reformation and SHFAs and their effect on shock erosion is to be investigated, it is important to model both features using realistic length scales. Other outstanding questions related to cavitons and SHFAs include why some cavitons develop into SHFAs and others do not, and

whether some SHFAs can survive/evolve downstream of the bow shock.

Although observations of the foreshock, bow shock and magnetosheath are abundant, it is statistically difficult to quantify from observations how close to the bow shock SHFAs form. According to simulations (Omidi et al., 2013), they form very close to the bow shock. Global simulations have specific advantages over point-like observations, providing large statistics and an easy way to disentangle spatial and temporal variations.

Foreshocks and their transient structures such as cavitons and SHFAs are ubiquitous features upstream of quasi-parallel shocks, and can therefore be found in other planetary environments in our solar system. In particular, SHFAs have recently been observed and modeled in the foreshock upstream of Venus and Mars (Collinson et al., 2017; Omidi et al., 2017). These works show that the size and properties of SHFAs, as well as their formation mechanism, are similar to that at Earth.

In this paper we perform a numerical study on the evolution and properties of cavitons and SHFAs based on simulations using

the strong capabilities of the Vlasiator hybrid-Vlasov code. Vlasiator facilitates a global simulation view while maintaining





realistic length scales and including ion kinetic physics, allowing us to present a statistical study on caviton and SHFA sizes. In particular, we show in detail how large SHFAs survive downstream of the bow shock and induce both the formation of a magnetosheath cavity and weakening and erosion of the bow shock.

The paper is organized as follows: In Section 2 we describe the Vlasiator code and the new data presentation methods to
identify cavitons and SHFA from simulations. The results are presented in Section 3 and discussed in Section 4.

## 2   Methods

Vlasiator (von Alfthan et al., 2014; Palmroth et al., 2015; Pfau-Kempf, 2016) is a unique hybrid-Vlasov code capable of performing global simulations of the Earth's magnetosphere and the surrounding space environment. It models kinetic proton-scale physics by simulating the proton distribution function through a cartesian 3-D velocity grid and cell-averaged values,
instead of relying on particle-in-cell methods and statistical sampling. Thus, Vlasiator has the inherent merit of being noiseless. A sparse velocity grid implementation maintains scalability and numerical efficiency.

Vlasiator models protons as a distribution function, solving the Vlasov equation for the ion (proton) distribution and closure being provided via Ampère's and Faraday's laws, as well as Ohm's law complemented by the Hall term. Electrons are modelled as a charge-neutralizing fluid, and due to a realistic proton mass and charge, kinetic effects are simulated on physical instead of
renormalized scales. As shown in Pfau-Kempf et al. (2018), kinetic proton phenomena are successfully reproduced even when the ion inertial ranges are not resolved, though spatial resolution does limit gradients, steepenings and thus possibly amplitudes of phenomena.

### 2.1   Vlasiator simulation run

In our investigation we use a global magnetospheric simulation performed in the meridional $(x - z)$ plane. The simulation
is 2-D in real space and 3-D in velocity space. In order to treat $y$-directional velocities self-consistently, periodic boundary conditions are employed at the $\pm y$ spatial cell walls. The solar wind has an $x$-directional inflow speed of $u_{\mathrm{sw}} = -750\,\mathrm{km\,s^{-1}}$ and the IMF magnitude is 5 nT, oriented towards the Sun and southward at a $45°$ angle. The solar wind number density is $n_{\mathrm{p}} = 10^6\,\mathrm{m^{-3}}$ and an ion temperature of $T = 0.5\,\mathrm{MK}$. The solar wind inflow conditions are steady throughout the run. We perform calculations for a total simulation time of 1437 seconds.

To be able to model foreshock features and interactions, our simulation box is extended from near-Earth space in the direction of the foreshock, with a box extent of $-48.6\,\mathrm{R_E}$ to $64.3\,\mathrm{R_E}$ in the $x$-direction and $-59.6\,\mathrm{R_E}$ to $39.2\,\mathrm{R_E}$ in the $z$-direction. The spatial cells are 300 km (1.3 solar wind ion inertial lengths) cubed, and our velocity resolution is set to $30\,\mathrm{km\,s^{-1}}$ (0.33 times the solar wind ion thermal speed) extending in all directions to $\pm 4020\,\mathrm{km\,s^{-1}}$. The polar setup (in the noon-midnight meridian plane) includes Earth's geomagnetic field as a line dipole, neglecting tilt, and with the dipole magnitude selected as outlined
in Daldorff et al. (2014) in order to result in a realistic magnetopause standoff distance. The inner boundary, set at 30 000 km $(\sim 4.7\,\mathrm{R_E})$ is modelled as a static Maxwellian, perfectly conducting ionosphere.





In order to more accurately and efficiently model the foreshock region, including regions where plasma density is decreased, we employ a sparse velocity space algorithm (see von Alfthan et al. 2014 and Kempf et al. 2015), that is, velocity space cells are dynamically allocated or discarded when their value is above or below a given threshold, respectively. The sparsity threshold value is scaled dynamically in accordance with proton number density. The minimum sparsity threshold is set at $10^{-17} \, \mathrm{m^{-6} \, s^3}$,

scaling up to $10^{-15} \, \mathrm{m^{-6} \, s^3}$ as a linear function of proton density between densities of $n_0 = 10^4 \, \mathrm{m^{-3}}$ and $n_1 = 10^5 \, \mathrm{m^{-3}}$.

We wrote reduced simulation output data to disk for data analysis with a simulation time interval of 0.5 seconds, in addition to retaining a full-state save at a time of 1187.85 seconds. The reduced data set contains magnetic and electric field components at cell boundaries, the Hall term for electric fields, the total number density and bulk velocity of protons and the pressure tensor. Additionally, the number density, bulk velocity and pressure tensor are provided for a subsection of the

distribution function, which is named the suprathermal beam population. This beam population is defined for this simulation as all protons not included in the core solar wind population, which in turn is considered to have a maximum thermal speed of $v_{\mathrm{core,max}} = 500 \, \mathrm{km \, s^{-1}}$ around the bulk velocity of $u_{\mathrm{sw,x}} = -750 \, \mathrm{km \, s^{-1}}$. We note that the beam population is well defined only upstream of the bow shock. The reduced data set additionally includes the proton distributions on a limited grid, every 50th cell in the $x$- and $z$-directions.

An overview of the simulation at $t = 1350 \, \mathrm{s}$ is shown in Figure 1. The colour code indicates the value of the magnetic field $B_y$ component, out of the plane of the simulation, and the black lines correspond to magnetic field lines in the $x$-$z$ plane. Regions inside the magnetopause are omitted from the figure so that only the regions of interest for the present study, the foreshock, the shock and the magnetosheath, are highlighted. Due to the IMF orientation, the foreshock develops in front of the southern part of the bow shock, as evidenced by the fluctuations of $B_y$ in this region. The oscillations of $B_y$ from positive to

negative values and the coherent wave fronts extending perpendicular to the IMF direction show that the foreshock is permeated by so-called 30 s ULF waves (Eastwood et al., 2005), with properties similar to those analyzed by Palmroth et al. (2015) in another Vlasiator run. Wave activity is also visible throughout the magnetosheath, with stronger perturbations downstream of the quasi-parallel portion of the bow shock. When the IMF has a significant southward component, reconnection takes place at the dayside magnetopause and creates magnetic islands which propagate tailward. One such island can be seen around

$x = 2 \, \mathrm{R_E}, z = -8 \, \mathrm{R_E}$. Reconnection in this run is studied in detail in Hoilijoki et al. (2018, in preparation).

## 2.2    Identification of structures

In this paper, we are focusing on two types of foreshock transients, cavitons and SHFAs. As discussed in the introduction, these structures are closely linked with each other, and it is believed that cavitons evolve into SHFAs. Cavitons are characterized by decreases of both the magnetic field strength and the density relative to the ambient plasma. In order to automatically detect

these structures in our simulation run, quantitative thresholds are set on these parameters. Following Kajdič et al. (2013) and Kajdič et al. (2017), we identify as cavitons those structures where the density and magnetic field strength are less than 80% of the solar wind density and magnetic field magnitude.

SHFAs are also characterized by decrements in density and magnetic field strength, but have in addition a higher temperature than the surrounding plasma. However, setting a criterion on the temperature is not straightforward since SHFAs are immersed

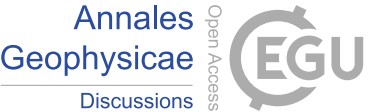



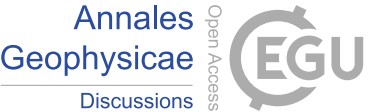

**Figure 1.** Zoom in to the magnetosheath and foreshock region of the Vlasiator simulation used in this analysis. The figure shows the region of solar wind interaction with the magnetosphere, with the colour scale indicating the out-of-plane magnetic field component $B_y$, showing fluctuations both in the magnetosheath and in the foreshock. Black lines indicate magnetic field lines in the $x - z$ plane. The inner boundary is placed at $5\,\mathrm{R_E}$, and the tail region is excluded from the plot for clarity.



in the foreshock, which has a higher temperature than the pristine solar wind. Similar challenges are encountered for the decrease in velocity associated with SHFAs, as deviations from the bulk solar wind velocity are observed throughout the foreshock, and they are not prominent enough inside SHFAs to be unambiguously identified. On the other hand, the enhanced temperature and the reduced magnetic field result in a significant increase of the plasma $\beta$. For this reason, we have set a

condition not on the temperature but on the plasma $\beta$ in order to identify SHFAs. We chose to use a threshold of $\beta > 10$, as it is significantly above the usual foreshock values.

## 3 Results

In this section we report on the analysis of cavitons and SHFAs using our global hybrid-Vlasov simulation data. At the beginning of our analysis period, we find a well-formed bow shock which does not display significant weakenings or transient

effects beyond minor shock rippling. Using the criteria for cavitons and SHFAs defined in Section 2.3, we identify these structures in the simulation and are able to track their evolution in time. At a given time in the run, about 50 cavitons and SHFAs are observed in the 2-D cut plane through the foreshock which we model in our run. In a 3-D run, the total number of these structures in the whole foreshock would most likely be larger. We will first concentrate on the formation of a magnetosheath cavity. We will then investigate the size distributions of cavitons and SHFAs in the foreshock. Finally, we will compare our

simulation results with spacecraft observations.

### 3.1 Formation of a magnetosheath cavity

Figure 2 shows three snapshots of the density of suprathermal beam population ions (panels b-d) and the magnetic field strength (panels e-g) within a $20 \times 20\,\mathrm{R_E}$ region covering a portion of the bow shock and foreshock. The region is initially centred on a position where multiple large cavitons form. Note that the position of the region of interest is moved along the bow shock

as time progresses, as illustrated in panel a, because the structures are convected by the magnetosheath flow. Black and green contours on the plots indicate structures fulfilling our requirements for a caviton and an SHFA, respectively (see Section 2.2). The blue contour marks where the density is equal to twice that of the incoming solar wind and is a good approximation of the bow shock position.

Time $T1 = 860\,\mathrm{s}$ features a structure that is evolving from a caviton (black contour around $x = 9\,\mathrm{R_E}$, $z = -18\,\mathrm{R_E}$ in panels

b and e) into an SHFA (green contour inside the black contour). As the caviton moves adjacent to the bow shock, it is filled with higher suprathermal ion density, causing it to evolve into an SHFA (Omidi et al., 2013). Further in the simulation more cavitons are seen approaching the bow shock and transforming into SHFAs. Time $T2 = 1105\,\mathrm{s}$ shows the phase when, as the result of SHFAs surviving crossing the bow shock, a large magnetosheath cavity has formed (yellow area around $x = -4\,\mathrm{R_E}$, $z = -33\,\mathrm{R_E}$ in panel f). "Chains" of SFHAs and cavitons are seen also upstream of the bow shock that add to the large

magnetosheath cavity. An indentation in the bow shock shape is developing where several large SHFAs have crossed into the magnetosheath, as evidenced by the blue contour in panels c and f. Time $T3 = 1350\,\mathrm{s}$ (close to the end of the simulation run) shows that the notch at $T2$ has turned into a large-scale weakening of the bow shock which extends deep within the





magnetosheath. At all times, and throughout the foreshock, we note that as in panels b-g, SHFA formation occurs closer to the shock whereas cavitons are generated further out.

To examine in more detail how the structure of the bow shock is modified by SHFAs, Figure 3 displays profile cuts in the spatial $x$-$z$ plane spanning a distance of $15\,\mathrm{R_E}$ from the magnetosheath into the foreshock. We plot six panels, showcasing

the proton number density $n_\mathrm{P}$, the suprathermal beam number density $n_\mathrm{P,beam}$, the magnetic field magnitude $|B|$, the ion temperature $T$, the plasma $\beta$, and the bulk flow speed $|V|$. We draw profiles at the three simulation times presented in Figure 2, that is $T1 = 860\,\mathrm{s}$, $T2 = 1105\,\mathrm{s}$, and $T3 = 1350\,\mathrm{s}$. Each profile is chosen to cut across those features of the shock which evolve into the large magnetosheath cavity. Cut extents were chosen so that the increase in $|B|$ corresponding to the shock position is located at the same position for all cuts. The cut positions at each time are shown as grey lines in panels b-d of Figure 2.

Rectangular shaded areas indicate those spatial regions which fulfill our caviton criteria of $n_\mathrm{P} < 0.8 n_\mathrm{P,sw}$ and $|B| < 0.8 |B_\mathrm{SW}|$, with the colour indicating the time at which the caviton is identified (gray, yellow, blue for T1, T2, and T3).

As the cuts and the feature of interest move in time along the shock front, originating in a region closer to the nose of the bow shock and propagating tailward and southward, we see a decrease in downstream proton density. At time $T2$ the jump in density across the shock is weakened, reaching barely values of twice the solar wind density. The overall lack of

clear density enhancement at time $T3$ indicates that the shock (positioned at approximately $x = 6\,\mathrm{R_E}$) has eroded away. The proton beam density at time $T1$ decreases strongly with distance from the shock, but as time progresses the extended beam further out strengthens and the profile flattens. It is noteworthy that times $T2$ and $T3$ show little difference in the beam density, indicating the reflection and isotropisation process of beam ions has reached a quasi-steady state. In the third panel of Figure 3, we see a strong indication of the formation of the magnetosheath cavity in the decrease in magnetic field strength at the

region $x < 6\,\mathrm{R_E}$. At time $T1$, we see a steep enhancement in $|B|$ at the shock location, with wave-associated periodicity in the downstream region. At time $T2$, there is still a weak peak at the shock position but the magnetosheath cavity has only a weak magnetic field due to heating and expansion. By time $T3$, the peak of $|B|$ at the shock position has eroded almost completely away, in agreement with our observations from proton density, indicating that the shock has eroded away. The fourth and fifth panels show that there is strong heating and a rise in plasma $\beta$ at the shock early in the simulation. Oscillations of temperature

in the magnetosheath smooth out by time $T2$, and in the region immediately downstream of the shock (i.e., in the cavity) the temperature remains somewhat constant, particularly at time $T3$. At the start of the cut, i.e., deeper in the magnetosheath, temperature decreases over time, which may be associated with the cut being further from the nose of the shock at time $T3$ than at earlier times. Finally, in the sixth panel of Figure 3, we see the evolution of bulk velocity over time, where an increase in the downstream bulk speed as a function of time can be attributed to the region of interest moving further away from the

nose, allowing for increased magnetosheath flow. To some extent, the bulk flow speed changes can also be due to a weakened shock being less efficient at decelerating the upstream plasma as it crosses the shock.

The shaded areas in Figure 3 show how at time $T2$ there are multiple large cavitons upstream of the shock. SHFAs are found at shaded areas where also $\beta > 10$. There is only a single SHFA at time $T1$, and there are only two small cavitons at time $T3$. The two SHFAs at time $T2$ exhibit decrements in $|B|$ and $n_\mathrm{P}$ of around $40 - 50\,\%$ from solar wind values and enhanced $\beta$

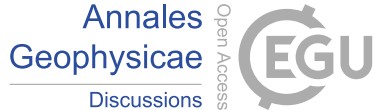



**Figure 2.** Evolution of cavitons, SHFAs, and a magnetosheath cavity in the Vlasiator simulation run. To track the evolution of the same structures, panels (b-d) correspond to a fixed dimension $20 \times 20\,\mathrm{R_E}$ box at three separate locations and time steps, as illustrated in panel (a). The times are: $T1 = 860\,\mathrm{s}$ (Region $\mathrm{R_1}$), $T2 = 1105\,\mathrm{s}$ (Region $\mathrm{R_2}$), and $T3 = 1350\,\mathrm{s}$ (Region $\mathrm{R_3}$). Panels (b-d) and (e-g) correspond to suprathermal ion density and magnetic field magnitude, respectively. The contours represent the bow shock (blue), cavitons (black), and SHFAs (green). The criteria for identifying each of these is provided in the legend, located in the top right corner. The three grey lines are positions of profiles chosen for further study.



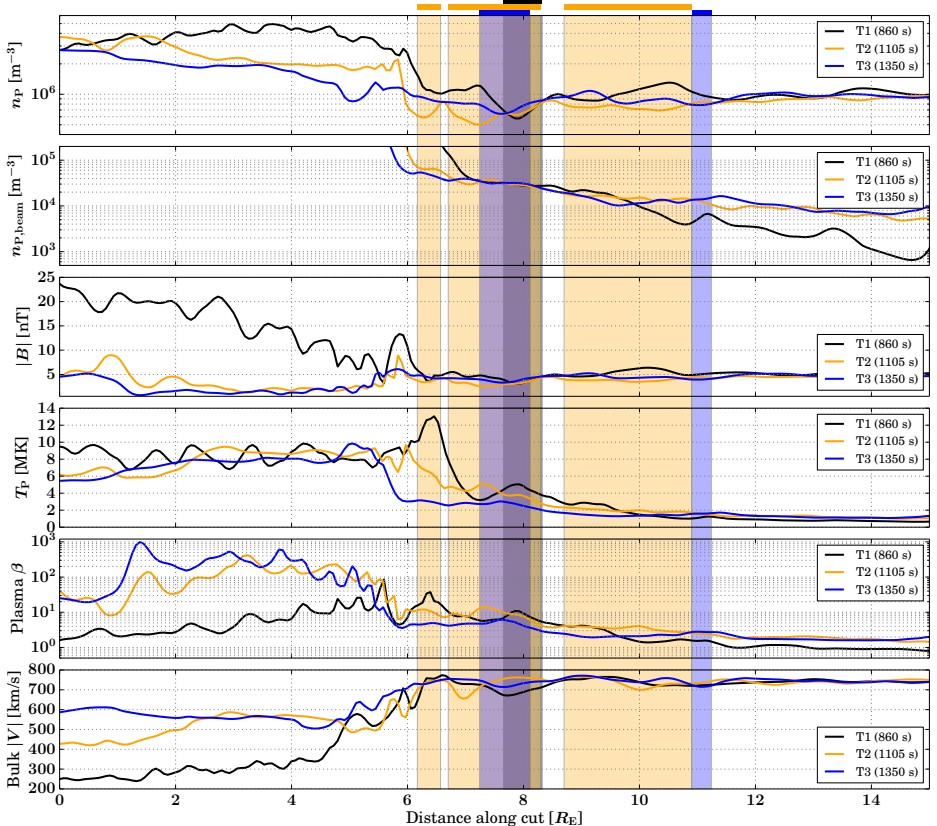

**Figure 3.** Profile cuts from the magnetosheath across the shock into the foreshock region at three different times of the simulation. The locations of the cuts were shown in Figure 2. Rectangular areas shaded in grey, yellow, and blue indicate the regions at each time ($T1$, $T2$, and $T3$, respectively) which fulfill our caviton criteria of $n_P < 0.8 n_{P,sw}$ and $|B| < 0.8 |B_{SW}|$. Each shaded area is accompanied by a indicator bar above the first panel. The beam density $n_{P,beam}$ is not well defined in the sheath and is thus allowed to saturate.

just exceeding a value of 10. In the sixth panel, small dips in the value of $|V|$ associated with cavitons and SHFAs can also be identified.

A hypothetical explanation for the lack of cavitons and SHFAs at time $T3$ might be a negative feedback process, where cavitons/SHFAs erode the shock, but once the shock has eroded, generation of cavitons would be suppressed as the weak shock allows for plasma to distribute more freely back to the upstream, filling the forming cavitons. At time $T2$ there are significant dips in proton density corresponding with the cavitons, whereas at time $T3$ the dips are less prominent. Alternatively, an eroded shock might result in smaller backstreaming beams and weaker upstream wave formation, suppressing caviton formation mechanisms associated with waves.



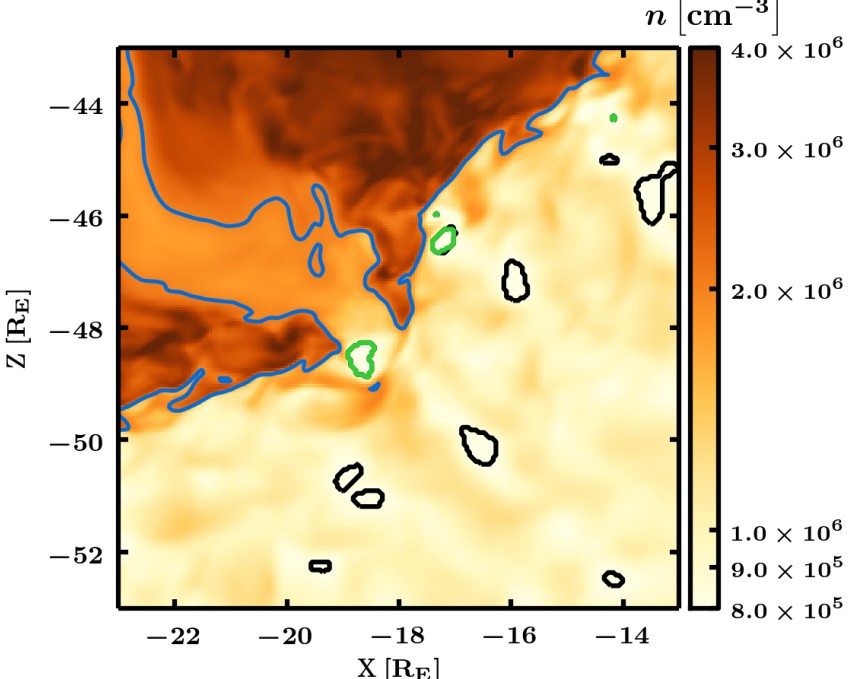

**Figure 4.** The local region around the magnetosheath cavity, showing the extent of bow shock erosion at time $T3 = 1350\,\mathrm{s}$. The figure extents are $10 \times 10\,\mathrm{R_E}$ with the colour scale showing proton number density on a logarithmic scale. The contours represent the bow shock (blue), cavitons (black), and SHFAs (green).

## 3.2   Shock erosion

Figure 4 shows (at time $T3$) a close-up of the magnetosheath cavity and of the weakened bow shock. The total ion density is indicated with the main colour scheme. The contours delineate again cavitons (black), SHFAs (green) and the bow shock (blue). The outstretched light orange region extending deep into the magnetosheath contains very weakly compressed plasma, as its

5   density is less than twice that of the inflowing solar wind. This density value corresponds to $\sim 50\%$ the value of the surrounding magnetosheath, and is in agreement with values observed inside magnetosheath cavities (Katırcıoğlu et al., 2009). The bow shock, as marked by twice the undisturbed solar wind density, has disappeared between $x = -19$ and $-18\,\mathrm{R_E}$. The eroded region, visible already at time $T2$, grows progressively as the magnetosheath cavity is convected tailward. A Supplementary Animation of images identical to Figure 4, following the features along the bow shock from region R1 to regions R2 and finally

10   R3 (see Figure 2), is provided online. The animation time extent is from 750 s to 1437 s, starting before time $T_1$ and continuing after time $T_3$. The growth of the magnetosheath cavity is most likely due to multiple SHFAs crossing the same part of the bow shock in rapid succession, augmented by the overall weakening of the bow shock when moving further from its nose. At the end of the simulation the magnetosheath cavity has grown in size to encompass a length of $> 5\,\mathrm{R_E}$ and a width of $1 - 2\,\mathrm{R_E}$. The rippling of the shock surface, seen throughout the quasi-parallel bow shock, causes local changes in shock geometry with





possible consequences for the formation of magnetosheath structures such as jets (Hietala et al. 2009, Palmroth et al. 2018, in preparation). Any such effects may be modulated by both SHFAs impinging on the shock and erosion such as seen associated with this magnetosheath cavity. Changes to shock properties will also alter the dynamics of the local foreshock and will affect reflection of particles and generation of foreshock ULF waves.

We note that although the plasma density within the cavity is low and the compression ratio becomes small compared to the undisturbed solar wind, the simultaneous decrease in magnetic field strength results in an increase in Alfvénic Mach number at this position, rising above values of 10 and peaking at values $> 40$. Other positions where multiple SHFAs cross into the downstream region also result in regions of high Alfvénic Mach number within the magnetosheath, but not right at the shock front. Magnetosonic Mach numbers in the sheath mostly grow with increasing distance from the nose of the shock,

but the cavity and bulge (with a low density and high temperature) have consistently lower magnetosonic Mach numbers than surrounding areas. Within the sheath, we found values of $M_{\mathrm{ms}} \leq 3$, and within the magnetosheath cavity, values of $M_{\mathrm{ms}} \leq 2$. If we consider only the shock-normal upstream bulk flow in calculating the magnetosonic Mach number, the number drops to $\sim 1$, which is evident as the shock bulges out towards the upstream at this location.

### 3.3   Spatial comparison of features

Investigation of SHFAs crossing the bow shock, building up a magnetosheath cavity and eroding the bow shock requires analysis of the development of features over time, as seen in previous sections, and also comparison of similar regions which result in different behaviour, located at different spatial positions.

In order to study proton velocity distributions at optimal locations, we examine the simulation at time $T_{\mathrm{R}} = 1187.85\,\mathrm{s}$, when we have a full-state save of the simulation data. Figure 5 shows a zoom-in to the region of interest, plotting the magnetic field

strength using the main colour scheme. The large magnetosheath cavity is visible as a large pale region, with other signs of shock deformation such as smaller magnetosheath rarefications visible further along the shock front. The plot shows contours for the shock front (blue), upstream cavitons (black), and upstream SHFAs (green), along with three profile cuts intersecting regions of interest and 15 positions selected for further study.

The first cut in Figure 5, labelled 1, starts from within the large magnetosheath cavity, crosses the eroded shock front and

extends into the foreshock. Positions A and B are within the magnetosheath cavity, with A situated well within the structure and B located close to the shock within the bulge extending beyond the regular shock position. Position C is located within an upstream SHFA, D is further out within an upstream caviton, and E is located even further along the same field lines at a position without significant features.

Cuts 2 and 3 of Figure 5 are located further along the shock front, parallel to upstream field lines, with cut 2 along a foreshock

path which sees limited caviton formation, and cut 3 along a path where caviton formation is significant. At cut 3, SHFAs are weakened during their crossing into the downstream, the shock is not eroded, and a strong magnetosheath cavity does not form. Positions F through K and L through P are positioned similarly starting from within magnetosheath features, proceeding across the shock, located at upstream cavitons and finally far in the upstream in regions lacking significant features.

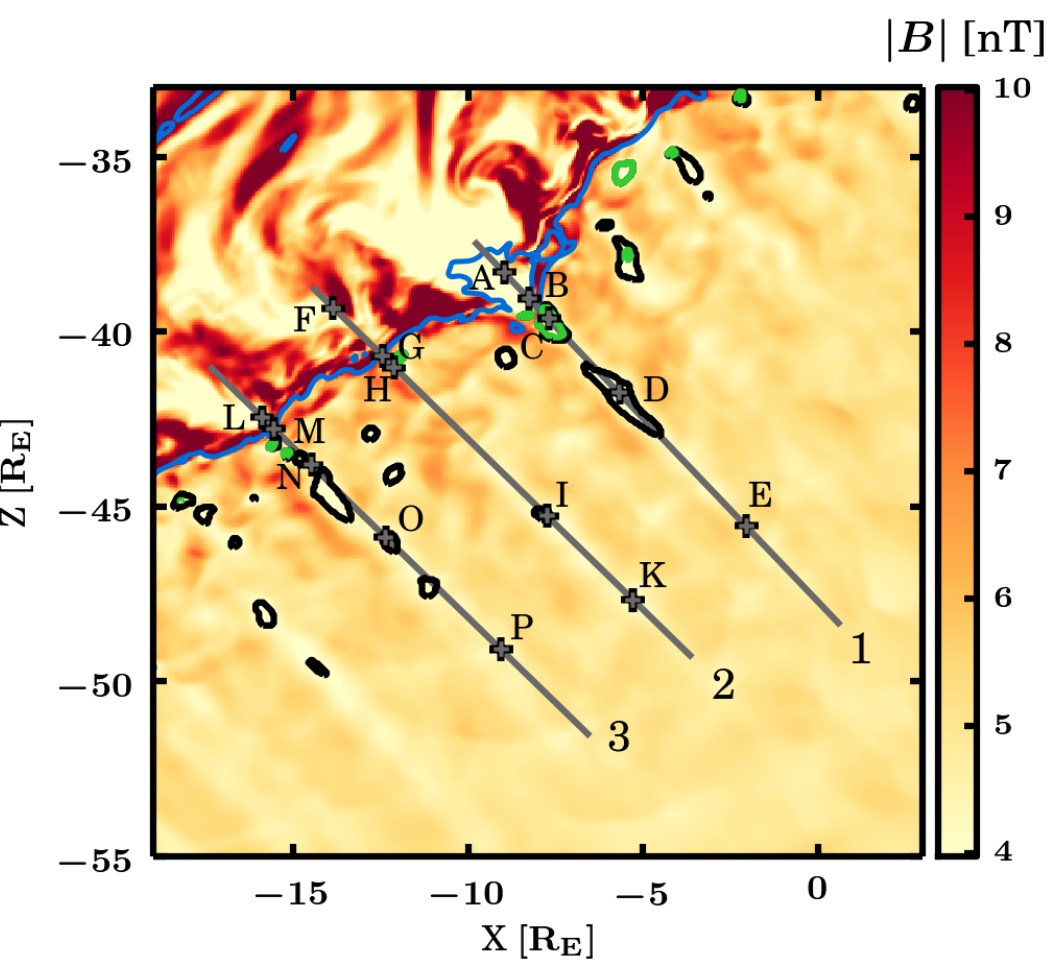

**Figure 5.** A plot showing the magnetic field strength at time $T_R = 1187.85$ s. The contours represent the bow shock (blue), cavitons (black), and SHFAs (green). The lines labelled 1, 2, and 3 are the positions of profile cuts through three regions of interest. The letters and adjacent crossmarks indicate positions where we examine proton velocity distributions.



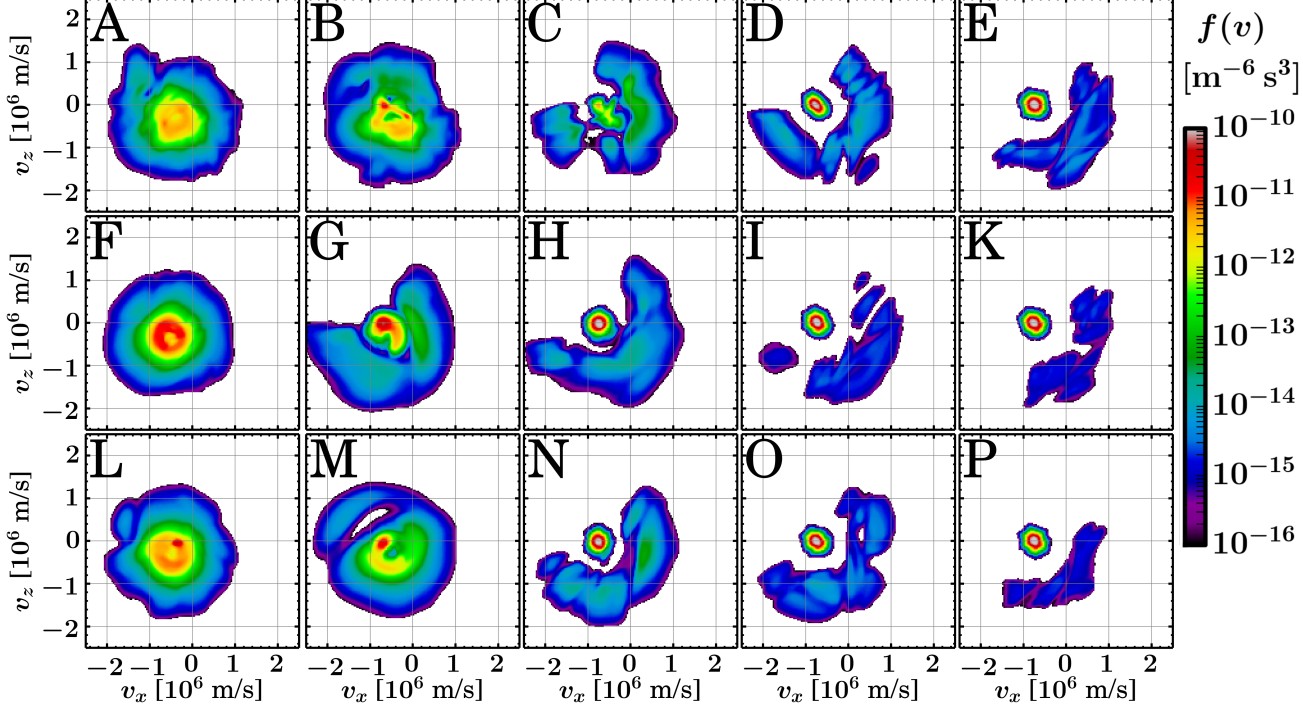

**Figure 6.** Cuts of the proton velocity distribution function at time $t = 1187.85$ in the $v_x$-$v_z$ plane at fifteen positions (labelled A–P). The positions and the cuts they were chosen from are shown in Figure 5. Each row corresponds to one cut, with panels from left to right going from the magnetosheath to the bow shock, then an SHFA, then a caviton further out, and finally an upstream position without transient features.

In Figure 6, we display $v_x$-$v_z$-directional cuts of 15 proton velocity distribution functions (VDFs), labelled A–P, matching the positions shown in Figure 5. These VDFs showcase samples of how the core solar wind population is modified across the shock and how the beam population evolves as a function of distance from the shock. We note that our proton VDFs are fully three-dimensional and these panels, showing $v_x$-$v_z$-directional cuts, exclude features of the VDF which would be visible only in the $v_y$-directional component. Due to the gyromotion of charged particles, we consider the plots as representative of the total upstream VDFs. Panels A, F, and L show a heated quasi-isotropic sheath population, although panel A shows decreased density and an additional weak field-aligned downstream beam. Panels B, G, and M show the magnetosheath population being formed, with the core being heated and deformed. Panel B, within the bulge of the magnetosheath cavity, shows a strongly deformed, asymmetric population. Panel G shows a rotationally deforming and stretching core along with an upstream beam which is on the way towards becoming isotropised. Panel M also shows a rotationally stretched and heated solar wind core, but the beam population has already merged with it significantly. It also appears that panel M displays a distinct downstream beam. The third, fourth, and fifth columns show how with increasing distance from the shock, the reflected ion beam strength decreases and how the extent of beam particle pitch-angles is limited to a narrow field-aligned beam for the ions observed far upstream.



Panel C, within an upstream SHFA, shows a strongly deformed and depleted core with almost all-encompassing beam pitch-angles, indicating that the vicinity of the shock bulge has a strong effect on the SHFA VDF. We would also like to draw the reader's attention to the fourth column, where the strength of the beam corresponds with the strength of features – panel D (cut 1) has largest beam intensities, followed by panel O (cut 3), with panel I (cut 2) having only a weak beam. Although the cavity

associated with panel I is further away from the shock as those associated with panels D and O, investigation of nearby VDFs suggests that distance is not a dominating parameter here. In the final column, panel E, associated with the cut connected to the eroded region of the shock shows also a stronger beam than panels K or P.

Figure 7 shows spatial profiles, similar to Figure 3, but all at time $T_{\mathrm{R}} = 1187.85\,\mathrm{s}$ over the cut locations shown in Figure 5. Cut positions were chosen so that the increase in $|B|$ corresponding to the shock front is located at the same position for

all cuts. In addition to the six panels similar to those shown in Figure 3, we show, for each cut, a panel with a heat map of proton pitch-angle cosine $\cos(\alpha)$ distribution, for the suprathermal beam portion of particles (as defined in section 2.1). Due to the beam being well defined only in the foreshock region, we allow the maps to saturate in the magnetosheath. We also plot rectangular areas shaded in grey, yellow, and blue to indicate the regions at each of the cuts (1, 2, and 3, respectively) which fulfill our caviton criteria of $n_{\mathrm{P}} < 0.8 n_{\mathrm{P,sw}}$ and $|B| < 0.8|B_{\mathrm{SW}}|$.

In the first panel of Figure 7 we see the striking decrease in proton number density associated with cut 1, which crosses the magnetosheath cavity, both in the magnetosheath and in regions of the foreshock. The low proton density also means that large regions in the foreshock easily fulfill the density requirement of our caviton criteria, falling even below 50% of the mean solar wind density. In agreement with panels (c) and (d) of Figure 2, there is little difference between the three cuts in beam density, as seen in the second panel of Figure 7. This is due to all three cuts being located along fingers of enhanced foreshock

beam density, where there is abundant reflection of particles and associated formation of cavitons. In the gaps between these fingers, beam densities are lower, and formation rates of cavitons and SHFAs are much lower. These fingers are convected along with the solar wind as are the corresponding bow shock features. We note that due to the large variation of beam density with distance from the shock, and thus, the requirement to plot the beam density on a logarithmic scale, variations associated with cavities and SHFAs appear less pronounced. The magnetosheath cavity at cut 1 is also strongly visible in magnetic field

magnitude (third panel), with $|B|$ peaking at the shock front position but falling to sub-foreshock levels in the sheath. Cuts 2 and 3 show more varied fluctuations in $|B|$, though all cuts show a peak at the shock position.

In the foreshock, SHFAs at cut 1 show magnetic field decreases of up to 50% from the solar wind magnetic field. Cut 1 shows significant heating, both in the magnetosheath cavity and in the foreshock SHFA visible right in front of the shock, whereas the temperature profiles of cuts 2 and 3 (fourth panel) do not differ from each other much. In plasma $\beta$ (fifth panel), cut 1 shows

SHFA-related enhancements, but cuts 2 and 3 also show slightly lesser enhancements in the magnetosheath. In bulk velocity (sixth panel), we see a shock-associated dip in all three cuts, with magnetosheath values rising with distance from the shock nose, in agreement with what was seen in panel 6 of Figure 3.

In the lower three panels of Figure 7, we display heat maps of the pitch-angle distributions (PADs) for the suprathermal beam portion of protons, as measured in the rest frame of the plasma. The shaded rectangular regions highlighting cavitons extend to

cover these maps as well. We begin by noting how for cut 3, the blue caviton regions match increased PAD spread fairly well,





**Figure 7.** Profile cuts from the magnetosheath across the shock into the foreshock region at three different positions (as shown in Figure 5), at time $T_R = 1187.85\,\mathrm{s}$. Rectangular areas shaded in grey, yellow, and blue indicate the regions at each of the cuts (1, 2, and 3, respectively) which fulfill our caviton criteria of $n_P < 0.8 n_{P,sw}$ and $|B| < 0.8|B_{SW}|$. Each shaded area is accompanied by a indicator bar above the first panel. The bottom three panels display heat maps of pitch-angle cosine $\cos(\alpha)$ for the suprathermal beam distribution of protons, measured in the plasma reference frame.



with an especially wide spread at the shockward edge of the caviton. For cut 2, with the single yellow region, there is only a weak match in PAD spread and caviton location. There are multiple locations along cut 2 where the PAD extends beyond $-0.5$, such as at $r \sim 5\,\mathrm{R_E}$, and two peaks near $r \sim 6\,\mathrm{R_E}$. It is important to note that at those locations, there are dips in proton number density and magnetic field strength, resembling cavitons. However, due to this cut being in a region of the foreshock

where ambient values for $n_\mathrm{P}$ and $|B|$ are enhanced, the dips, though being large enough to signify caviton formation, no longer match our caviton criteria, which were chosen based on mean upstream solar wind values. In a similar but opposite fashion, at cut 1 we see a giant caviton further away from the shock, but a strong signature in the PAD is seen only at the centre of it, at $r = 6\,\mathrm{R_E}$. Thus, if we assume that the PAD spread is a signature of a caviton, meaningful caviton criteria for each cut should be based on the local, not global solar wind values. The low plasma density found at cut 1 may, however, be a rare occurrence,

and the strong magnetic field at cut 2 may be a result of heated expansion of plasma at cuts 1 and 3 causing the field at cut 2, between them, to be enhanced.

In summary, we have identified and tracked the formation and the evolution of a transient structure which seeds a significant disturbance at the shock and in the magnetosheath. This structure originates from large SHFAs, which have evolved from cavitons when approaching the bow shock, due to the increasing density of suprathermal ions. When crossing the bow shock,

the large SHFAs create a region of enhanced temperature, decreased plasma density and weakened magnetic field, with an associated increase in plasma $\beta$. This downstream region, which grows as it is convected tailward along the shock, was termed a *magnetosheath cavity*, following the terminology introduced in Katırcıoğlu et al. (2009) and Omidi et al. (2016) for a similar phenomenon. This structure is strengthened subsequently by additional SHFAs which cross the shock at the same position. The portion of the bow shock adjacent to the magnetosheath cavity erodes away progressively, reducing significantly the jump

in magnetic field strength and plasma density observed at this position. As the magnetosheath cavity plasma becomes tenuous, the core solar wind VDF is broken down and heated in an uneven and anisotropic manner. Other SHFAs of similar size as the one presented here are observed in other parts of the foreshock and crossing the bow shock, but none of them lead to such a significant erosion of the bow shock. In the foreshock, we find features which resemble cavitons and SHFAs at many different positions, but conclude that due to local enhancements and depletions in plasma density and magnetic field intensity, global

caviton and SHFA criteria fail to accurately capture all facets of foreshock transients. We suggest that suprathermal beam ion PADs are an useful tool for identifying SHFAs in the foreshock.

### 3.4 Size distributions of cavitons and SHFAs

The global view of the foreshock provided by the simulation does not only allow us to track the evolution of specific structures as they convect past the bow shock, but also to look at the evolution and some of the properties of cavitons and SHFAs

throughout the foreshock. As evidenced by the black and green contours in the Supplementary Animation, SHFAs (in green) are only found within a few $\mathrm{R_E}$ from the bow shock, whereas cavitons can appear much further out. This is consistent with the fact that the density of suprathermal ions is larger in the vicinity of the bow shock, as is visible for example in panels b–d of Figure 2, thus resulting in an enhanced temperature and plasma $\beta$ which fulfill our SHFA criteria. More importantly, we note that most cavitons evolve into SHFAs when approaching the bow shock (black contours becoming green in the animation),





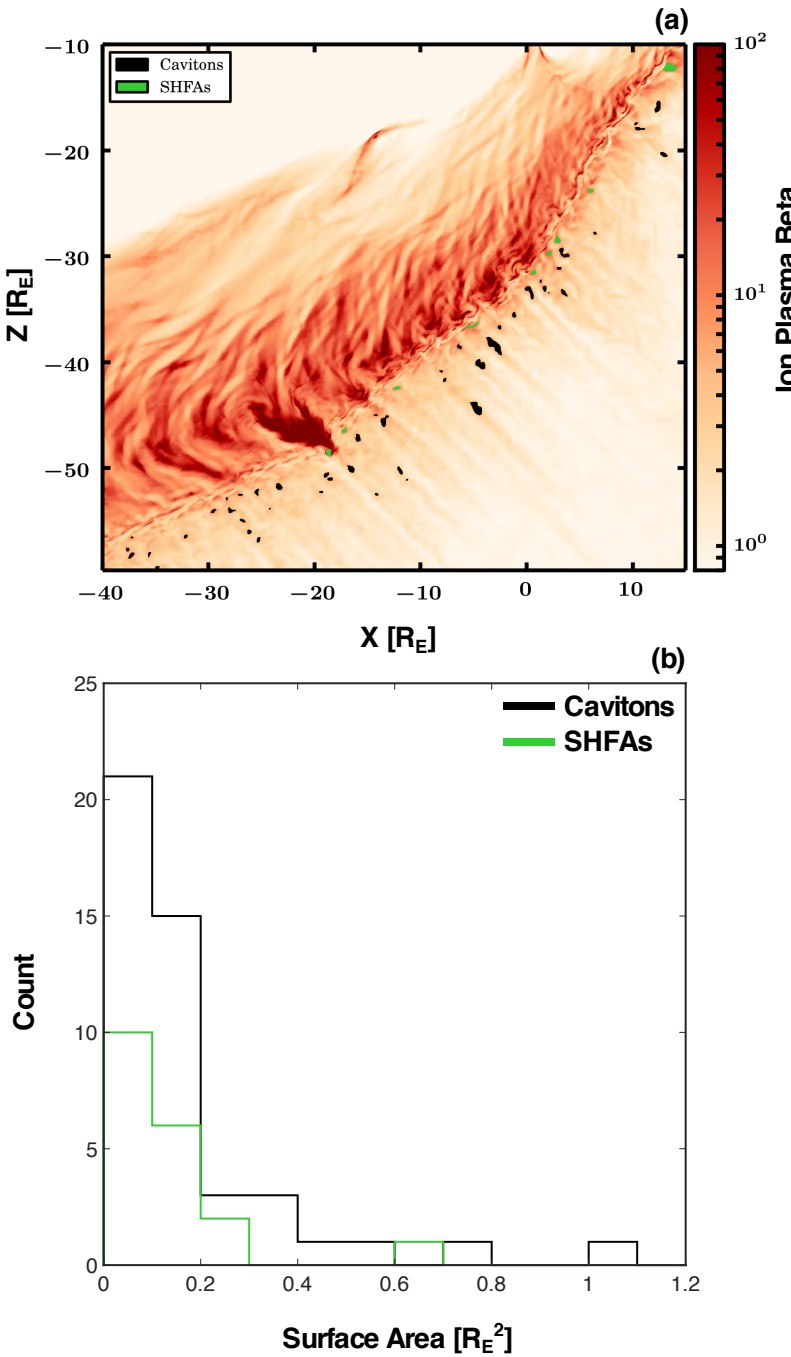

**Figure 8.** Spatial distribution of cavitons and SHFAs and their associated physical scales in terms of surface area. Panel (a) shows the global spatial distribution of cavitons and SHFAs at $T3 = 1350\,\mathrm{s}$. Cavitons and SHFAs can be identified from the black and green regions, respectively. The colour scale of the simulation run corresponds to ion plasma $\beta$ on a logarithmic scale. Panel (b) represents an histogram of the caviton and SHFA surface areas in $\mathrm{R_E^2}$. The colour of cavitons and SHFAs is the same as panel (a).





in agreement with earlier simulations by Omidi et al. (2013), who first suggested that SHFAs originate from the interaction of cavitons with suprathermal ions coming from the bow shock. Cavitons that do not evolve into SHFAs disappear before impinging on the shock, with very few exceptions, while some SHFAs are born as such, without initially being identified as cavitons. These two phenomena may both be due to the limitations of our criteria for automated structure detection. The

thresholds we have selected for the magnetic field intensity and the plasma density are both based on the mean upstream solar wind values, whereas these parameters can vary across the foreshock. For example, in regions of enhanced plasma density, cavitons may appear with density greater than 80% of that of the solar wind, but still be significantly depleted compared to the ambient plasma. Ideally, the thresholds should be set based on the average local plasma parameters, but this is not applicable on a global scale.

We then investigate the size distribution of cavitons and SHFAs in the total simulated foreshock region. Note that in this part of the analysis we need to identify structures either as cavitons or SHFAs, without overlap between them. Therefore, for each caviton, we check which fraction of it has a plasma $\beta$ above 10. If 60% or more of the caviton area has $\beta > 10$, then the entire structure is considered as an SHFA. We then calculate the surface area of each of the structures. Figure 8a shows the cavitons (in black) and SHFAs (in green) identified at $T3 = 1350\,\mathrm{s}$, which is representative of other times in the run, overplotted on a

colour map of the plasma $\beta$. The dark red feature around $x = -20\,\mathrm{R_E}$ corresponds to the magnetosheath cavity we discussed earlier. In total, 46 cavitons and 19 SHFAs are detected at this time. An histogram of their surface area is displayed in Fig. 8b for both types of structures, with the same colour code as before. Both distributions peak close to zero, showing that the smallest structures are the most numerous, while larger structures are rarer. The shape of the distributions of both types of structures are very similar, which supports the hypothesis that cavitons evolve into SHFAs.

Though cavitons are much smaller than other foreshock structures such as for example foreshock bubbles (Archer et al., 2015), some of them reach sizes up to 1 $\mathrm{R_E^2}$, which can be large enough to cause local disturbances at the bow shock. The magnetosheath cavity which was analysed in detail in the previous sections was initially triggered by SHFAs of about 1 $\mathrm{R_E^2}$, thus showing that these structures may in some cases have large-scale effects, leading for example to the shock erosion shown in Figure 4. We note however that our simulation includes other structures of similar size which do not have such a strong

impact on the bow shock.

### 3.5    Comparison with spacecraft observations

Figure 9 shows a time series from a virtual spacecraft positioned at $x = 0\,\mathrm{R_E}$, $z = -35\,\mathrm{R_E}$ around the time when a caviton, marked by the green area on the plot, crosses this location. The black dashed lines in the two upper panels indicate the undisturbed solar wind values of the magnetic field strength and the ion density, while the red dashed lines correspond to

our identification criteria for a caviton. Both criteria are fulfilled in the core of the structure. Even though the plasma $\beta$ (blue dashed line in third panel) is lower than the limit we set in Section 2.2 to define an SHFA, it is clear that this caviton is evolving towards an SHFA, as the $\beta$ is already much higher than in the surrounding plasma. It also contains a significantly larger density of suprathermal ions (bottom panel), showing that particles have started to accumulate inside the structure, and thus causing the increase in temperature and $\beta$.



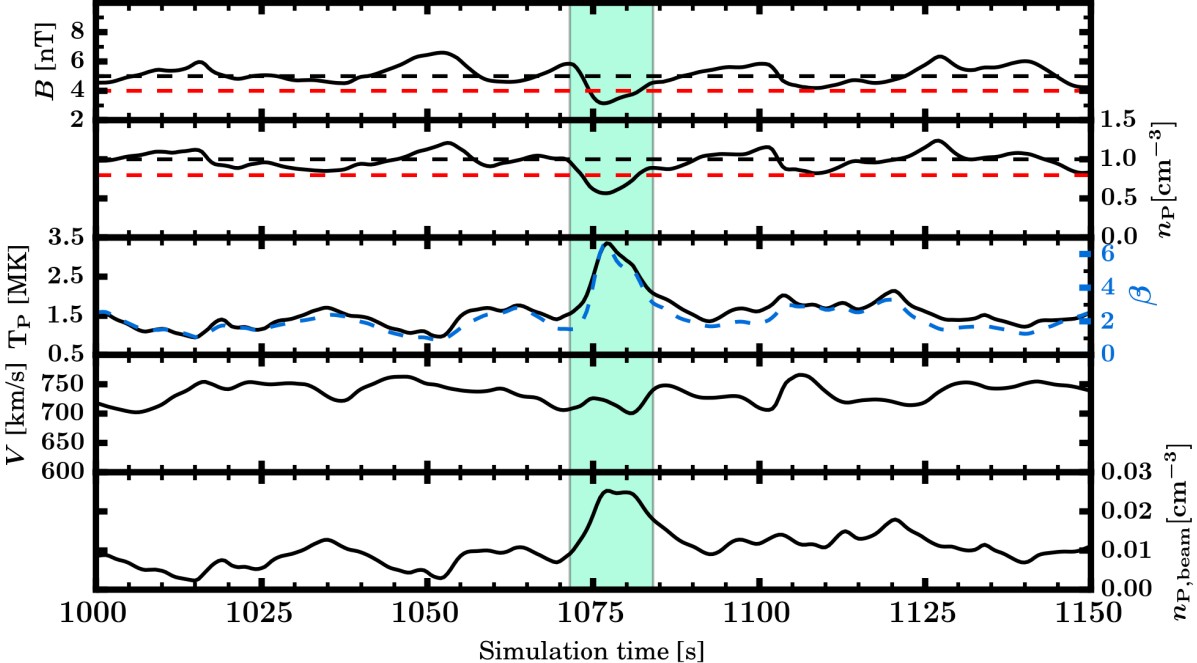

**Figure 9.** Time series of a virtual spacecraft data extracted from the simulation domain at $x = 0\,\mathrm{R_E}$, $z = -35\,\mathrm{R_E}$ from $t = 1000\,\mathrm{s}$ to $t = 1150\,\mathrm{s}$. The panels show, from top to bottom: the magnetic field magnitude, the proton density, the proton temperature (black) and the plasma $\beta$ (blue), the solar wind bulk velocity and the beam proton density.

The profile of the caviton in Fig. 9 can be directly compared to cavitons observed in the Earth's foreshock by the Cluster spacecraft (Kajdič et al., 2011, 2013). In particular, Figure 3 of Kajdič et al. (2011) shows an example of a caviton for similar interplanetary magnetic field strength and ion density as in our simulation. The decrements of these two parameters are slightly more pronounced in the spacecraft data, down to 2.5 nT and 0.3 $\mathrm{cm^{-3}}$, and the gradients at the edges of the cavitons are

sharper, but the structure resembles closely the caviton showcased in Fig. 9. The flux of energetic particles measured by the spacecraft inside the cavitons is roughly doubled, which corresponds well to the enhanced density of suprathermal ions shown in the bottom panel of Fig. 9. In the simulation, the waves surrounding the caviton show some compression in agreement with observations that have demonstrated that cavitons are immersed in a sea of compressive waves. ULF waves similar to these have been reported in Palmroth et al. (2015) which also uses Vlasiator data. We note however that compressive waves in this

Vlasiator run have amplitudes in the magnetic field magnitude and density of around 10-20 % the background values. These amplitudes are smaller than what is observed at Earth near cavitons, which is typically 50% of the average magnetic field (Kajdič et al., 2010). Compressive waves in the foreshock can have even larger amplitudes closer to the shock (Hoppe et al., 1981). This may be due to the spatial resolution in our simulation, which can limit the steepening of the waves (Pfau-Kempf et al., 2018). This may also explain why the caviton shown in Fig. 9 does not display the "shoulders" of enhanced plasma

density identified in spacecraft observations on either sides of the density and magnetic field depression.



In the central part of the structure, the magnetic field strength and plasma density decrease to about 50% of their solar wind values, as is the case on average for cavitons, according to a statistical study performed by Kajdič et al. (2017). Only the duration of the structure, which is about $25\,\mathrm{s}$ here, does not match so well with previous works and is at the lower end of the distribution obtained by Kajdič et al. (2017). This is due to a combination of two factors. One is that the solar wind

velocity in our run is $750\,\mathrm{km\,s^{-1}}$, which corresponds to conditions during a high-speed stream or a solar wind transient, thus causing the caviton to convect faster past the virtual spacecraft than during regular solar wind conditions. For average solar wind speed (about $400\,\mathrm{km\,s^{-1}}$), the same caviton would last about 46 s in the virtual spacecraft time series. Another factor that can affect the observed duration of the structure is how we define where the edges of the cavitons are. Specifically, the "shoulders" surrounding the cavitons are included in the caviton durations in Kajdič et al. (2017), but this cannot be done in our

simulation, as these features are not as markedly defined. Overall, our results show a good agreement with the observations, both qualitatively and quantitatively.

## 4   Conclusions

In this paper we have used the hybrid-Vlasov simulation software Vlasiator to study foreshock transients and their effect on the bow shock of the Earth. Our main foci have been cavitons and SHFAs and their transition into a magnetosheath cavity.

Vlasiator simulations confirm previous experimental and simulation results that cavitons evolve into SHFAs as they approach the bow shock and fill up with a high density of suprathermal ions, which have been reflected by the bow shock. Vlasiator improves on previous studies both in modelling the realistic scales of all transient structures, and in providing high-quality noise-free velocity distribution functions throughout the simulation.

The primary result of this study is that large SHFAs can survive downstream of the bow shock and erode it, creating a large-

scale structure of low density, high $\beta$ plasma extending deep into the magnetosheath. In our simulation, the bow shock erosion seems to be initially triggered by SHFAs spanning about 1 $\mathrm{R_E^2}$, which are among the largest observed during the run. Though other SHFAs of comparable size also appear elsewhere in the foreshock, only one magnetosheath cavity forms. Our analysis suggests that this is likely due to the fact that numerous SHFAs impinged the shock roughly at the same place. In particular, our simulation run shows that chains of cavitons and SHFAs form along a narrow, roughly field-aligned, band of decreased plasma

density (after $t = 1050\,\mathrm{s}$ in yellow in the Supplementary Animation) and successively hit the bow shock. These subsequent cavitons and SHFAs may contribute to the growth of the magnetosheath cavity and the shock erosion. The higher density of suprathermal ions and associated higher temperature causes a pressure increase in this narrow region, which may explain the decreased total ion density and magnetic field. We also propose that a crucial factor in facilitating this is that the initial large SHFA crosses the bow shock rather close to the nose, where the bulk flow velocity in the magnetosheath parallel to the

shock front is relatively small. Therefore, the forming magnetosheath cavity is convected quite slowly along the bow shock, with more time to grow, whereas further downstream, the SHFAs which have crossed the shock propagate away from the nose inside the magnetosheath **faster** and thus are not strengthened by other structures, or at least not to such a large extent. This effect is self-strengthening, as bulk flow velocities within the magnetosheath cavity are lower than those in adjacent parts of



the magnetosheath. The heating of the magnetosheath cavity leads to a decrease in magnetosonic Mach number $M_{\mathrm{ms}}$, which results in the cavity-associated region of the bow shock bulging out into the incoming solar wind. Dynamics of SHFA-triggered erosion of other planetary bow shocks may be different due to different magnetic field strengths and spatial scales. The fact that multiple SHFAs are needed for the formation of large magnetosheath cavities is in agreement with the results of Omidi

et al. (2016).

It is interesting to note how the spatial distribution of the cavitons and SHFAs in the foreshock changes with time. As can be seen in Figure 2e-g, the chains of cavitons and SHFAs along field-aligned bands of decreased plasma density and magnetic field strength are only visible at $T2$ and $T3$, but not at $T1$. One of these bands of tenuous plasma and weaker magnetic field is associated with the magnetosheath cavity, but others are observed at other places along the bow shock, thus ruling out the

fact that these bands are solely the result of some feedback of the magnetosheath cavity on the upstream medium. On the other hand, we note that the magnetosheath density is lower downstream of these structures, thus suggesting that they may feed the magnetosheath cavity by causing an additional density decrease. Disentangling in which ways the upstream and downstream media, and the bow shock itself, influence each other and control the spatial distribution of the foreshock transients is however left for future work.

One of the properties associated with SHFAs is that of decreased bulk velocity, and our simulation did display dips at the locations of cavitons and SHFAs. However, the decrease seen in our data is much less than that reported by Omidi et al. (2013), Zhang et al. (2013), and Zhao et al. (2015). To investigate this more, we tracked the changes visible in proton distribution functions as foreshock features were convected across virtual spacecraft, and found that our VDFs resembled those found by Zhang et al. (2013). That is, the flow of the thermal solar wind core was not slowed or deflected, but rather, changes in bulk

flow are due to the combination of a density decrease for the core and a strengthening of the suprathermal beam. When the thermal core is depleted, the backstreaming beam can have a relatively greater impact on bulk velocity measurements.

Even though one would intuitively think that there would be a lack of suprathermal ions upstream of a weaker portion of the bow shock, we find that the suprathermal bean density does not vanish, as shown by the orange "fingers" extending into the white region in Figure 2c. The profiles shown in Figure 7 show that suprathermal densities of three adjacent caviton-generating

fingers are roughly similar, despite one of them being in front of the eroded portion of the shock. The bow shock is strongly distorted in the vicinity of the magnetosheath cavity, which changes the local $\theta_{B\mathrm{n}}$ and can therefore affect the amount of backstreaming particles in this area. It is also possible that the magnetosheath cavity acts as a heated ion reservoir, so that ions can leak into the upstream medium and populate the foreshock, leading to extra SHFA formation in this region. Finally, the decreased plasma density at the cavity leads to an increase in Alfvénic Mach number. This would result in any Alfvénic

fluctuations convected from the upstream to the downstream to pile up within the magnetosheath cavity. These fluctuations could potentially act as an efficient scattering barrier for resonant energetic ions, enabling them to be reflected back into the upstream, despite the shock having a low density compression ratio and a low magnetosonic Mach number in the vicinity of these features.

Upstream of the cavity and the eroded bow shock, we were able to examine VDFs of suprathermal protons in the plasma

frame, and found SHFA-associated increases in pitch-angle spreads. Beam particles had particularly large pitch-angle spreads



at the shockward edges of SHFAs. The cavity and eroded shock caused strong deformation of the core solar wind population close to the shock. Regions associated with strong formation of cavitons and SHFAs tended to have stronger beam intensities.

We note that for both observational and simulational studies, care must be taken in defining caviton and SHFA criteria in order to prevent masking choices from influencing, e.g., size distributions. We observed local regions of plasma and magnetic
field enhancements or rarefactions, causing the chosen global selection criteria to preferentially detect large cavitons in regions of tenuous plasma and small cavitons in regions of dense plasma. We observed that proton pitch-angle distribution widths have a correlation with cavitons and SHFAs, with the strongest spread at the shockward edge of cavitons. Thus, we recommend using proton VDFs as an additional selection criteria. Determining the spatial and/or temporal extent of the foreshock to use as the background level for selection criteria remains an open question and is likely influenced by solar wind conditions. It
remains to be seen how much variation local background levels have in global 3-D hybrid-Vlasov simulations.

The large magnetosheath cavity that develops in this run has similar features to the magnetosheath cavities reported on by Omidi et al. (2016), with decreased values of magnetic field magnitude, ion density and high temperature. Observational work presented in Katırcıoğlu et al. (2009) shows the existence of such structures in the Earth's magnetosheath. These authors predicted that the existence of such structures represents a decrement in the total pressure applied to the magnetosphere and
can allow the magnetopause to move $30\%$ further from Earth, compared with the position predicted from the far upstream solar wind. For a HFA and an associated IMF tangential discontinuity, Sibeck et al. (1999) reported magnetopause movement on the order of $5\,\mathrm{R_E}$. Our simulations did not, however, indicate notable magnetopause movement in reaction to SHFAs or the magnetosheath cavity. This is likely due to the cavity growing to its full strength only as it has travelled further away from the nose of the shock. The fast solar wind velocity in our run may also play a role, as the structure convects relatively quickly
along the bow shock.

We also note that, as somewhat visible in Figure 8a, our simulation results in structures similar to magnetosheath filamentary structures (Omidi et al., 2014a), with the magnetosheath cavity strongly connected to a prominent filament. Caprioli and Spitkovsky (2013) reported on a cosmic-ray induced filamentary instability in a parallel shock. At certain phases of their hybrid simulation, a feature at their shock front bore a striking resemblance to our magnetosheath cavity, although filamentation with
included enhancements in magnetic field appeared to be the dominating feature instead of sheath heating. The connection between filaments and SHFA bow shock crossings would be a potential topic for further study. A detailed study of these connections is facilitated by the realistic sizes of both types of structures provided by Vlasiator modelling.

The dependence of, e.g., caviton formation on different IMF geometries (Blanco-Cano et al., 2011) is something our single simulation run cannot explore. Future possible extensions of this work would be the analysis of SHFA and caviton formation
rates and size distributions as well as shock erosion in relation to different Mach numbers, IMF conditions and solar wind parameters. The numerical requirements associated with global high-resolution hybrid-Vlasov modelling make parametric studies challenging, but not impossible.

Our results show that cavitons evolve into SHFAs only within a few Earth radii of the bow shock, but also that this evolution occurs at distances beyond those associated with SLAMS and shock reformation in these solar wind conditions. Previous
hybrid modelling investigating these phenomena (see, e.g. Omidi et al. 2013) has provided fascinating results, but our results



suggest that this connection between reformation, SHFA formation, magnetosheath cavity formation, and bow shock erosion should be carefully investigated using simulations with realistic physical length scales, such as those provided by Vlasiator, in order to distinguish between these different phenomena.

*Code and data availability.* Vlasiator is distributed viahttps://github.com/fmihpc/vlasiator under the GPL-2 open source license. This ad-
dress contains links to the analysator software used to produce the figures. The run described here takes several terabytes of disk space and is kept in storage maintained within the CSC – IT Center for Science. Vlasiator uses a data structure developed in-house (http://github.com/fmihpc/vlsv), which is compatible with the VisIt visualisation software (https://wci.llnl.gov/simulation/computer-codes/visit) with a plugin available at the above address. Data presented in this paper can be accessed by following the data policy on our web page http://www.physics.helsinki.fi/vlasiator/rules.php.

*Competing interests.* The authors declare that they have no conflict of interest.

*Acknowledgements.* This paper was outlined and drafted in the First International Vlasiator Science Hackathon held in Helsinki, 7-11 Aug 2017. The Hackathon was funded by the European Research Council Consolidator grant 682068 - PRESTISSIMO, awarded to MP.

We acknowledge The European Research Council for Starting grant 200141-QuESpace, with which Vlasiator (http://www.physics.helsinki.fi/vlasiator/) was developed, and Consolidator grant 682068-PRESTISSIMO awarded to further develop Vlasiator and use it for
scientific investigations. The Finnish Centre of Excellence in Research of Sustainable Space, funded through the Academy of Finland grant number 312351, supports Vlasiator development and science as well. We also gratefully acknowledge the Academy of Finland (grant numbers 267144, 267186, and 258963). The CSC – IT Center for Science in Finland is acknowledged for the Sisu supercomputer pilot usage and Grand Challenge award leading to the results presented here. X. Blanco- Cano acknowledges the UNAM PAPIIT-DGAPA (IN105014-3) and CONACyT (255203). The work of L. Turc was supported by a Marie Sklodowska-Curie Individual Fellowship (#704681). A. P. Dim-
mock was supported by the Swedish Civil Contingencies Agency, grant 2016-2102. E. Kilpua acknowledges Academy of Finland project 1310445. This project has received funding from the European Research Council (ERC) under the European Union's Horizon 2020 research and innovation programme (grant agreement n° 4100103).





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
