# Peer review of "Cavitons and spontaneous hot flow anomalies in a hybrid-Vlasov global magnetospheric simulation"

_Annales Geophysicae, 2018_

## Referee Comment (RC1) · Anonymous Referee #1 · 9 Apr 2018

This manuscript show us very detailed description of the foreshock cavitons and the formation of spontaneous hot flow anomalies (SHFAs), and stated their close relationship with a Vlasiator global magnetospheric hybrid-Vlasov simulation code. The simulation results are interesting, and will contribute to the study of quasi-parallel shocks. But, in this manuscript, some previous papers should be added into the references. Page 2-line 5: papers associated with geomagnetic disturbances should be added. Page 2-line 9: maybe a review of collisionless shock should be added. Page 3-2nd paragraph: you may refer to some simulation results from Scholer et al. [1993JGR] and recent simulations performed by Hao et al. [2016, JGR, doi:10.1002/2015JA021419; 2016, APJ, doi:10.3847/0004-637X/823/1/7; 2017, JGR, doi:10.1002/2017JA024234]. After

the revision, I think the manuscript would be suitable for publication in ANGEO.

---

## Referee Comment (RC2) · Anonymous Referee #2 · 12 Apr 2018

The manuscript investigates properties of the ion foreshock, in particular cavitons and spontaneous hot flow anomalies using a 2D global hybrid-Vlasov simulation. The authors present new interesting new results, the work presents an incremental advance, the presented results confirm previous results and include more detailed analysis of cavitons and spontaneous hot flow anomalies and their effects on the shock and the magnetosheath. The manuscript is acceptable for publications in AG after some relatively minor corrections, see below:

The manuscript needs a small reorganisation, presentation of results and discussions ought to be well separated.

[Figure]

Note that foreshock is not the same thing as the upstream region. Usually, electron/ion foreshocks are distinguished; these are regions where there exists populations of electrons/ions reflected off the shock. The region magnetically connected to the bow shock is almost the same as the electron foreshock but the ion one is rather different.

What the upstream ion/electron betas? What are the resulting Mach numbers? It is necessary to include basic plasma/shock parameters in theoretical units.

"... and due to a realistic proton mass and charge, kinetic effects are simulated on physical instead of renormalized scales" What does it mean?

"The spatial cells are 300 km (1.3 solar wind ion inertial lengths) cubed ..." cubed?

The grid size is relatively large (generally larger than the local ion inertial length and likely also larger than the ion gyroradius). This may strongly affect the shock structure; the typical size of the shock front is rather a small fraction of the on inertial length.

"As shown in Pfau-Kempf et al. (2018), kinetic proton phenomena are successfully reproduced even when the ion inertial ranges are not resolved, though spatial resolution does limit gradients, steepenings and thus possibly amplitudes of phenomena." This belongs to section 4. Note that this claim is quite questionable: It is more or less clear that only some kinetic proton phenomena could be successfully reproduced even when the ion inertial ranges are not resolved. Some phenomena are strongly affected or even inhibited by an insufficient resolution (eg. generation of ion whistler waves). Also this reference to the unpublished manuscript of Pfau-Kempf et al is not sufficient, it is necessary to include some summary of their results.

It would be useful to discuss effects of the global structure (shock curvature). It is not clear if only the movement along the shock front is important, a change of the reference frame just remove it. With respect to the results of Omidi et al (2012) the size of the magnetosphere is relatistic, is it? The observed size of cavitons and SHFAs is relatively small with respect to the global scale of the shock. This may justify a local approach

that may allow a better resolution and extension to (spatial) 3D.

Other limitations of the work (2D geometry) needs to be discussed. "In a 3-D run, the total number of these structures in the whole foreshock would most likely be larger." is too speculative.

Figures: it may be better in some cases to show relative values (e.g., with respect to their upstream values) of different quantities or denote the upstream values in color scales. It would be also useful to use in some cases (eg. Figs. 3, 7, 9) "theoretical" units (c/omega_pi, Omega_i) along with the "physical" ones in order to facilitate comparisons of the presented results with results of previous numerical simulations (eg. Omidi et al 2013). Also a table with parameters of the simulation in physical as well as theoretical units would be useful.

Figs. 3 and 7 are quite confusing (especially the shaded areas); it may be better to separate the three different times/cuts.

Fig. 6: At what position in v_y these cuts are taken?

Fig. 7: Is the pitch angle calculated with respect to the local magnetic field?

---

## Author Comment (AC1) · 17 May 2018

We thank the referee for his/her suggestions. We have included most of the suggested references.

Page 2-line 5: papers associated with geomagnetic disturbances should be added.

Done, the text now appears in page 2 line 23, we added Eastwood et al., 2015, Archer et al., 2013

> Page 2-lines 8-9: maybe a review of collisionless shock should be added. Done, we added the references of Bale et al., 2005, and Burgess et al., 2005.

[Figure]

>Page 3-2nd paragraph: you may refer to some simulation results from Scholer et al. [1993JGR] and recent simulations performed by Hao et al. [2016, JGR, doi:10.1002/2015JA021419; 2016, APJ, doi:10.3847/0004-637X/823/1/7; 2017, JGR, doi:10.1002/2017JA024234].

We added Hao et al., 2017 on page 3 third paragraph, in relation to shock rippling, and Scholer et al. (1993) related to shock heating, structure, and processes. We added a reference to Hao et al., JGR 2016 about jets on page 11, line 34. The Hao 2016 ApJ paper is also an interesting treatise, but more focused on the response of specific ions to a reforming shock front - we will cite this in our similar studies of ion reflection at the bow shock.

---

## Author Comment (AC2) · 18 May 2018

> The manuscript needs a small reorganisation, presentation of results and discussions ought to be well separated.

A citation to Katircioglu et al. 2009 and Omidi et al., 2016 has been added on page 8, line 17, to keep part of the information given in one of the removed text, see next comment.

The last paragraph of section 3.3 was removed, as the contents are also included in the Conclusions.

[Figure]

The last paragraph of 3.5 was cut, as the content is included on page 22, lines 8-13

> Note that foreshock is not the same thing as the upstream region. Usually, elec-tron/ion foreshocks are distinguished; these are regions where there exists populations of electrons/ions reflected off the shock. The region magnetically connected to the bow shock is almost the same as the electron foreshock but the ion one is rather different.

We have elaborated on the sections discussion the foreshock, clarifying that we are discussing the section upstream of the quasi-parallel region. We agree that the electron foreshock is completely different, being connected to the quasi-perpendicular region.

> What the upstream ion/electron betas? What are the resulting Mach numbers? It is necessary to include basic plasma/shock parameters in theoretical units.

We have added the clarification that our electrons are modelled as a cold fluid, and thus a separate electron beta cannot be calculated. We added theoretical parameters derived from upstream input conditions, but wish to emphasize that the corresponding values within the foreshock, at the shock, and in the magnetosheath vary due to plasma interactions.

> "... and due to a realistic proton mass and charge, kinetic effects are simulated on physical instead of renormalized scales" What does it mean?

We elaborated that normalized scales referred to ion scales (i.e. ion gyroperiod $\Omega_{cp}^{-1}$ and ion skin depth $c\omega_{pi}^{-1}$)

> "The spatial cells are 300 km (1.3 solar wind ion inertial lengths) cubed ..." cubed?

Each cell is a cube in shape, with an edge length of 300 km. We have reworded this.

> The grid size is relatively large (generally larger than the local ion inertial length and likely also larger than the ion gyroradius). This may strongly affect the shock structure; the typical size of the shock front is rather a small fraction of the on inertial length.

We agree that an increased spatial resolution would likely allow us to resolve the spatial
structures and gradients at the shock better. Comparison with the coarser resolution of von Alfthan et.al. (2014), however, suggests that the shock is modeled to a reasonable extent.

> "As shown in Pfau-Kempf et al. (2018), kinetic proton phenomena are successfully reproduced even when the ion inertial ranges are not resolved, though spatial resolution does limit gradients, steepenings and thus possibly amplitudes of phenomena." This belongs to section 4. Note that this claim is quite questionable: It is more or less clear that only some kinetic proton phenomena could be successfully reproduced even when the ion inertial ranges are not resolved. Some phenomena are strongly affected or even inhibited by an insufficient resolution (eg. generation of ion whistler waves). Also this reference to the unpublished manuscript of Pfau-Kempf et al is not sufficient, it is necessary to include some summary of their results.

We are happy to state that this paper has been accepted for publication,and the full-text will be shortly available at https://www.frontiersin.org/articles/10.3389/fphy.2018.00044/abstract

The Pfau-Kempf paper is specifically written to answer these questions, and is thus a description of the physical capabilities of the Vlasiator software. We respectfully suggest keeping this section here, as this text does not describe conclusions drawn from the results in this paper. We have reworded the sentence somewhat, acknowledging that high-frequency effects (such as ion whistler waves) cannot be accurately modeled at this resolution.

> It would be useful to discuss effects of the global structure (shock curvature). It is not clear if only the movement along the shock front is important, a change of the reference frame just remove it.

We do consider the possibility that a position close to the nose is required for strong shock deformation, and, as suggested in our conclusions, flow speed changes due to convection along the shock front are likely to affect the dynamics of shock erosion. We

have added some commentary to the introduction on the expected different phenomena at different portions of the bow shock.

> With respect to the results of Omidi et al (2012) the size of the magnetosphere is relatistic, is it? The observed size of cavitons and SHFAs is relatively small with respect to the global scale of the shock. This may justify a local approach that may allow a better resolution and extension to (spatial) 3D.

While Omidi et al (2013) list their magnetosphere as 5 times smaller than the Earth's magnetosphere, we indeed scale our line dipole to result in a realistic standoff distance for the magnetopause. In this run, the magnetopause reaches about $9R_E$. Thus, our magnetosphere is of a more realistic size. As for the sizes of the cavitons, spacecraft observations have so far not provided 3-D measurements, but the virtual spacecraft measurements shown in section 3.5 show that our results, taking into account our solar wind driving conditions and the question regarding measurement of caviton shoulders, are in agreement with observations. Thus, the global effect of cavitons and SHFAs on the shock should be simulated more accurately in our model. Whilst a local model might allow finer resolution and 3D, it could not find results like the prolonged shock erosion reported in our paper. Thus, each approach has merit.

> Other limitations of the work (2D geometry) needs to be discussed. "In a 3-D run, the total number of these structures in the whole foreshock would most likely be larger." is too speculative.

We removed the sentence in question. As we do not have a 3-D simulation of suitable fidelity to compare with, estimating more differences between 2D and 3D runs would be even more speculative.

> Figures: it may be better in some cases to show relative values (e.g., with respect to their upstream values) of different quantities or denote the upstream values in color scales.

Indicating values in physical units allows for better comparisons with observations than plotting them relative to upstream quantities. Additionally, giving results in relation to upstream quantities might give the false impression that the results of this paper would directly apply to other solar wind conditions, when in fact different Mach numbers and IMF configurations may alter results significantly.

We also note that in e.g. figures 3 and 7, all quantities tend to converge to their upstream values when approaching the right-hand side of the figure.

> It would be also useful to use in some cases (eg. Figs. 3, 7, 9) "theoretical" units $(c/omega_p i, Omega_i)$ along with the "physical" ones in order to facilitate comparisons of the presented results with results of previous numerical simulations (eg. Omidi et al 2013). Also a table with parameters of the simulation in physical as well as theoretical units would be useful.

To aid in comparison with existing simulations, we have included a list of theoretical units based on upstream initialisation parameters within the Vlasiator run simulation section. However, as plasma densities and magnetic fields vary greatly within the near-Earth region, such theoretical normalisations need by definition to be applied with some constant values, such as the solar wind inflow values. In our opinion, physical units are thus a better metric.

> Figs. 3 and 7 are quite confusing (especially the shaded areas); it may be better to separate the three different times/cuts.

Having three different times or cuts in a single panel allows for better comparisons between those values. In order to clarify the plots, we have decreased the opacity of the shaded regions.

> Fig. 6: At what position in $v_y the se cuts are taken?$

We clarified the caption to indicate they are taken at $v_y = 0$.

> Fig. 7: Is the pitch angle calculated with respect to the local magnetic field?

Yes. We have clarified the caption to include this.

---

## Author Comment (AC3) · 20 May 2018

Just to let the referee know, the paper by Pfau-Kempf et al. that we cite is now fully published at https://doi.org/10.3389/fphy.2018.00044.